# Evaluation of an Eight-Week Whole-Food Plant-Based Lifestyle Modification Program

**DOI:** 10.3390/nu11092068

**Published:** 2019-09-03

**Authors:** Erin K. Campbell, Mohammad Fidahusain, Thomas M. Campbell II

**Affiliations:** 1School of Medicine and Dentistry, University of Rochester, 601 Elmwood Ave, Rochester, NY 14642, USA; 2Concentra, Occupational Medicine, 1300 S. 320th St., Federal Way, WA 98003, USA

**Keywords:** plant-based diet, vegan diet, vegetarian diet, low-fat diet, weight loss, hypertension, hypercholesterolemia, hyperlipidemia

## Abstract

Poor diet quality is the leading cause of death both in the United States and worldwide, and the prevalence of obesity is at an all-time high and is projected to significantly worsen. Results from an eight-week group program utilizing an ad-libitum whole-food plant-based dietary pattern, were reviewed. There were 79 participants, all self-referred from the community, including 24 (30.4%) who were already vegetarian or vegan at baseline. Seventy-eight participants (98.7%) completed the eight-week program. Among completers, those with higher BMI at baseline lost a larger percentage of their body weight (total body weight loss of 3.0 ± SD 2.1%, 5.8 ± 2.8%, and 6.4 ± 2.5% for participants who had baseline BMI in normal, overweight, and obese range, respectively). The average weight loss for all the completers was 5.5 ± 3.0 kg (*p <* 0.0001). Final blood pressure and plasma lipids were reduced compared to baseline (SBP decreased 7.1 ± 15.5 mmHg (*p* = 0.0002), DBP decreased 7.3 ± 10.9 mmHg (*p <* 0.0001), total cholesterol decreased 25.2 ± 24.7 mg/dL (*p <* 0.0001), LDL decreased 15.3 ± 21.1 mg/dL (*p <* 0.0001)). Twenty-one (26.9%) participants were able to decrease or stop at least one chronic medication compared to two (2.6%) participants who required an increased dose of a chronic medication. Participants who were already vegetarian or vegan at baseline experienced statistically significant weight loss and reductions in total and LDL cholesterol. There was a non-significant trend toward less weight loss in these participants compared to participants who were non-vegetarian at baseline. Reductions in total and LDL cholesterol were not significantly different when comparing vegetarian or vegan and non-vegetarian participants. A whole-food plant-based dietary intervention may provide significant short-term benefits for both non-vegetarian, vegetarian, and vegan individuals.

## 1. Introduction

Poor diet quality is the leading actual cause of death in the United States, accounting for 529,299 deaths in 2016, with the majority of these deaths (83.9%) due to cardiovascular disease [1]. Aside from death, a high body mass index (defined as BMI >22.5) and diet are the second and third leading actual causes of disability after tobacco use [1]. Between 2015 and 2016, the obesity prevalence among adults in the United States was 39.8% [2], the highest prevalence recorded to date. A recently published simulation model predicted that 57.3% of today’s children will be obese by 35 years of age [3]. If this model holds true, the obesity epidemic, accompanied by its serious health consequences and high personal and societal costs, will be with us and worsening for decades to come.

The intervention we designed (T.C., E.C.), implemented (T.C., E.C), and evaluated (E.C., M.F.) is an intensive behavioral group approach utilizing a whole-food, plant-based diet. The dose (number of sessions) and approach (behavioral) of the intervention was influenced by the behavioral interventions for obesity reviewed by the United States Preventative Services Task Force [4]. The term “plant-based diet” can represent a relatively broad spectrum of dietary patterns which share a common feature of being comprised primarily of plants or components of plants. Common variations of dietary patterns containing more plant-based foods compared to typical American consumption include the Dietary Approaches to Stop Hypertension (DASH) diet, the Mediterranean diet, vegetarian and vegan diets, and a strict whole-food, plant-based diet that excludes entirely animal-based foods and most processed foods. Several of these diets have substantial scientific support of healthfulness. The Mediterranean diet, for example, has extensive support in both observational and intervention research detailing substantial benefit, particularly for cardiovascular disease but also other diseases as well [5].

We chose a whole-food, plant-based (WFPB) nutritional approach that strictly excluded animal-based foods and minimized processed foods, including all edible oils, for the intervention. This iteration of a plant-based diet was chosen based on evidence of the beneficial treatment effect of a low-fat, WFPB diet on a variety of chronic diseases. Interventions consisting of a whole-food, plant-based diet, alone or accompanied by other lifestyle changes, have demonstrated angiographic reversal of atherosclerotic lesions in ischemic heart disease [6,7], reduction in prostate specific antigen levels and less progression to treatment in men with low-grade prostate cancer choosing ‘watchful waiting’ [8], improved glycemic control in type 2 diabetes mellitus [9], and reductions in symptoms and inflammatory markers in rheumatoid arthritis [10,11]. The duration of these interventions ranged from 4 weeks [11] to one or more years (12 months [6,8], 13 months [10], 72 months [9], 5 years [7]). All but one [10] included a group education component. Mean weight loss in these interventions ranged from 3 to 5.76 kg. Improvements in total cholesterol and LDL cholesterol levels were measured in each intervention in which cholesterol was assessed [6,7,8,9].

Our goal for the design and implementation of this intervention was to establish a medically supervised behavioral intervention that would aid participants in weight loss and the reduction of cardiac risk factors by providing the knowledge and skills, peer and clinical support needed for the adoption of a WFPB diet. In this article, we reviewed the outcomes of this intervention in the form of a program evaluation.

## 2. Materials and Methods 

The University of Rochester Research Subjects Review Board determined that this program evaluation did not qualify as human subjects research, therefore requiring no additional board review or oversight (15 May 2018). 

### 2.1. Participants

There were no specific diagnoses required for or excluded from participation. Participants self-referred from Rochester, NY and the surrounding area, with initial participants learning of the program from community groups with overlapping interests (local vegan society and Ornish heart disease support group) and later participants by word of mouth and a local newspaper article on lifestyle modification programs. Program participation was self-pay.

### 2.2. Intervention

The intervention consisted of 3 individual medical visits and 16 group classes over an average of 8.9 weeks, totaling 33.75 h of contact. Participants enrolled and participated in classes in small cohorts of 9 to 16 participants. The intervention was repeated 7 times, each with a new cohort of participants.

Prior to beginning group classes, each participant met with the physician(s) (T.C., E.C.) for a medical visit to review his or her medical history, current medical issues, medications, and a 3-day food diary each participant recorded prior to the visit. Baseline weight, height, and blood pressure were measured at this visit. Brief individual medical visits also took place at the midpoint and completion of the 8 weeks of group education with weights and blood pressures measured at each visit. 

Group education consisted of 8 weeks of twice-weekly group classes, each 2 h in duration. A physician instructor (T.C. and/or E.C.) taught 14.5 of the 16 classes. A chef experienced in preparing meals consistent with the intervention diet taught a cooking demonstration (0.5 class) and a hands-on technique class (1 class). The curriculum was designed to provide participants with an understanding of the science supporting a plant-based dietary pattern, the practical knowledge and skills necessary to select ingredients for and prepare whole-food, plant-based meals, behavioral strategies for making these dietary changes, and the group support of peers. A WFPB catered meal was provided at each class; no other food was provided. The curriculum focused heavily on plant-based nutrition but also touched on evidence of health benefits of physical activity and mindfulness meditation. There was no mandated exercise or meditation component. The goal of the 8-week program was facilitating the long-term adoption of a WFPB dietary pattern. The dietary recommendations of the program were modelled after interventions demonstrating reversal of ischemic heart disease [6,7], which included minimal to no animal-based foods and exclusion of added oils and high-fat plant foods (avocados, nuts, and seeds). With the exception of a hands-on cooking class at a local teaching kitchen, all the classes took place at either a UR Medicine primary care office or a meeting room at the UR Medicine Center for Primary Care. Each participant had weight and blood pressure measured weekly. 

Participants were asked to adhere to a diet of fruits and non-starchy vegetables, cooked starchy plants, limited portions of high fat plant foods, and daily consumption of ground flax or whole chia seeds as an omega-3 fatty acid source (Table 1). They were asked to avoid all animal-based foods, refined grains and flours, and added oils, and to minimize added sugars. Added sugars were allowed in savory uses (as a minor ingredient in a salad dressing or sauce, for example), but participants were asked to avoid sweets and added sugars in beverages. A vitamin B12 supplement was recommended.

Participants were instructed to consume this diet ad libitum. Calorie counting was discouraged. Participants were not given a specific target for their fat consumption but instructed to strictly avoid added oils. Participants desiring more significant weight loss were encouraged to choose non-starchy vegetables and fruits for at least half of the volume of their food intake. They were encouraged to choose non-starchy vegetables and fruits for snacks over more energy dense foods like crackers, nuts, and seeds. Participants who did not want to lose weight, or wanted to gain weight, were encouraged to choose more high fat whole plant foods.

### 2.3. Anthropometric Measures

Height was measured with shoes removed using a wall-mounted stadiometer at the initial medical visit. Weights were measured using a calibrated digital medical scale (I Series, SR Scales by SR Instruments, Inc., Tonawanda, NY, USA). Participants removed outer clothing and the majority of participants removed shoes. Participants who insisted on wearing shoes had all of their weights performed wearing shoes. Weights were not adjusted for shoes but recorded as they appeared on the scale. Weights were recorded in pounds (the unit familiar to participants).

Blood pressures were measured manually for cohorts 1–4 or using a digital blood pressure monitor for cohorts 5–7. Manual blood pressures were obtained using DS66 sphygmomanometer and appropriately sized cuff (Welch Allyn, Skaneateles Falls, NY, USA). Automated blood pressures were obtained using one of two blood pressure monitors: either Series 10 Model BP785N (Omron, Lake Forest, IL, USA) with regular cuff or LifeSource UA-789 (A&D Company, San Jose, CA, USA) with bariatric cuff. Upper arm circumference was measured at the initial medical visit and the same monitor or sphygmomanometer and appropriately sized blood pressure cuff was used for each participant’s measurements.

### 2.4. Laboratory Measures

For cohorts 1–4, fasting lipid panels were ordered by the physician at the initial medical visit. These same labs were repeated at the end of the 8 weeks of group classes for the majority of participants (not all participants completed a final blood draw). For cohorts 5–7, point-of-care non-fasting lipid panels (Smart Bundle, PTS Diagnostics, Indianapolis, IN, USA) was performed using a point-of-care analyzer (CardioChek Plus, PTS Diagnostics, Indianapolis, IN, USA) for each participant at initial, mid-point, and program completion visits. Point-of-care testing was performed at the primary care office by one of the physicians.

### 2.5. Statistical Analysis

Paired t-tests were performed to determine whether participants’ measures at program completion demonstrated statistically significant differences from their baseline measurements. Unpaired t-tests were used to compare outcomes between participant subgroups. All t-tests were two-tailed. An ANOVA model was used to test differences in percent weight loss between normal weight, overweight, and obese participants from baseline to program completion. Chi-square test of independence analyses were performed to determine if participants who were vegetarian or vegan at baseline differed from participants who were non-vegetarian at baseline on any other baseline categorical variables. Pairwise deletion was used to address missing data.

Analyses were performed using SAS 9.4 (SAS, Cary, NC, USA).

## 3. Results

### 3.1. Participant Characteristics

Between August 2015 and July 2017, seventy-nine participants enrolled and 78 (98.7%) participants completed the 8-week intervention in seven consecutive cohorts. The one participant who dropped out of the program did not give a reason or respond to our attempts at contact. The mean age of participants was 59.2 ± 10.7 years (range 25–79 years). The majority of participants were female (67%, *n =* 53), white (85%, *n =* 67), and/or obese (61%, *n =* 48). Thirty percent (*n =* 24) of participants reported a vegetarian or vegan diet at enrollment (Table 2). Of note, there were no statistically significant differences in baseline BMI, systolic blood pressure, diastolic blood pressure, total cholesterol, HDL cholesterol, or LDL cholesterol when comparing vegetarian or vegan participants to non-vegetarian participants. 

### 3.2. BMI and Weight

Clinical outcomes are shown in Table 3. Program completers (*n =* 78) experienced a mean BMI reduction of 2.0 ± 1.1 kg/m^2^ (*p* < 0.0001). The mean weight reduction was 5.5 ± 3.0 kg (*p* < 0.0001) over an average of 8.9 weeks. This corresponds to a mean body weight reduction of 5.7%. Both overweight (5.8 ± 2.8%) and obese (6.4 ± 2.5%) participants lost a significantly (*p* < 0.05) greater percentage of body weight than normal weight (3.0 ± 2.1%) participants. Overweight and obese participant subgroups did not differ significantly from one another in percent body weight reduction (n.s.). Vegetarian or vegan participants lost a lower percentage of their body weight compared to non-vegetarian participants, but this difference did not reach statistical significance (Table 4). The chi-square analysis of independence did not demonstrate a relationship between baseline dietary pattern and BMI category.

### 3.3. Blood Pressure

Baseline and program completion data for systolic and diastolic blood pressure was available for 75 (96%) program completers. Both systolic (−7.1 ± 15.5 mm Hg, *p* = 0.0002) and diastolic blood pressure (−7.3 ± 10.9 mm Hg, *p* = 0.0001) decreased significantly. There were no significant differences in either systolic or diastolic blood pressure reductions between participants with or without a hypertension diagnosis. Likewise, there were no significant differences in blood pressure reductions between participants who did or did not require changes to antihypertensive medications (n.s.). Vegetarian or vegan participants experienced smaller, non-significant reductions in blood pressure when compared to non-vegetarian participants (Table 4). Chi-square analysis of independence did not demonstrate a relationship between baseline dietary pattern and hypertension diagnosis. 

### 3.4. Cholesterol and Triglycerides

Baseline and program completion cholesterol and triglyceride data were available for 65 (83%) program completers. Mean total cholesterol reduction was −25.2 ± 24.7 (*p* <0.0001). Mean total LDL cholesterol reduction was −15.3 ± 21.1 (*n =* 63; LDL could not be calculated by the point-of-care testing device for two participants due to elevated triglyceride levels, *p* < 0.0001). HDL cholesterol decreased by a mean of −7.3 ± 7.8 (*p* < 0.0001). Triglycerides were reduced (−10.2 ± 63.2), but this change was not statistically significant (n.s.). There were no statistically significant differences in lipid reductions for any of the cholesterol components or triglycerides when comparing participants taking statin medications with participants who were not taking statin medications. Likewise, there were no significant differences in lipid reductions between participants with or without a dyslipidemia diagnosis. Vegetarian or vegan participants experienced a smaller reduction in HDL cholesterol (−4.2 ± 7.0 mg/dL vs. −8.8 ± 8.0 mg/dL, *p* = 0.0336) when compared to non-vegetarian participants. Vegetarian or vegan participants triglycerides increased, but this was not statistically significant (15.3 ± 74.9 mg/dL, n.s.), while non-vegetarian participants’ triglycerides decreased (−19.6 ± 56.2 mg/dL, *p* = 0.0253). This difference between the groups was statistically significantly (*p* = 0.0456). There were no significant differences in reductions in total or LDL cholesterol when comparing vegetarian or vegan and non-vegetarian participants (Table 4).

### 3.5. Medication Changes

Twenty-one (26.9%) participants were able to decrease dosage or frequency or discontinue entirely at least one chronic prescription medication. Two (2.6%) participants required an increase in a chronic prescription medication. One participant had inadequately controlled type 2 diabetes mellitus and another had inadequately controlled hypertension at baseline, necessitating these medication increases. The major medication classes in which medications were decreased or discontinued were anti-hypertensives, anti-hyperglycemics, and gastroesphageal reflux disease treatments.

## 4. Discussion

### 4.1. Main Findings

Evaluation of this eight-week group education program utilizing an ad libitum whole-food, plant-based diet demonstrated statistically significant weight loss, reduction of systolic and diastolic blood pressure, and total and LDL cholesterol reductions. Over a quarter of participants were able to decrease or discontinue at least one chronic medication. The program attracted a substantial proportion (30%) of participants who identified as vegetarian or vegan prior to the intervention. These participants experienced statistically significant weight loss and total and LDL cholesterol reductions. There was a non-significant trend towards less weight loss in these participants when compared to participants who identified as non-vegetarian at baseline. Weight loss [6,7,8,9,10,11,12] and total and LDL cholesterol reductions [6,7,8,9,12] have been demonstrated in similar plant-based interventions. The reductions in blood pressure were consistent with evidence from controlled trials and observational studies associating vegetarian diets with lower systolic and diastolic blood pressure [13].

The degree of weight loss achieved in this intervention is notable given its short-term nature, ad libitum diet, and lack of mandated exercise. When taken together, participants with a BMI of 25 or greater experienced 6.2 ± 2.5% body weight loss. Participants in the lifestyle arm of the Diabetes Prevention Program, all of whom were also overweight or obese, achieved an average weight loss of 5.6 kg, an approximately 6% body weight loss, at the completion of the 24-month curriculum. At the earlier 6 and 12-month time points, average weight loss was approximately 7 kg or just over 7% body weight loss. This weight loss was achieved with calorie and fat restriction and mandated exercise [14]. A randomized controlled trial of a bi-weekly group education program utilizing a whole food plant-based diet to address obesity and cardiovascular risk factors published in 2017 demonstrated 9.1% body weight loss in 12 weeks in the WFPB intervention group [12]. The subjects in the study were overweight with type 2 diabetes mellitus, ischemic heart disease, hypertension, or hypercholesterolemia, or obese. As in our intervention, subjects were instructed to consume a WFPB diet ad libitum and there was no mandated exercise component. The intervention group results were particularly notable for a total of 12.8% body weight loss at 6 months (3 months post-intervention), which was largely maintained at 12 months (6 months post-intervention) with a total of 12.1% body weight loss. For some individuals, an ad libitum WFPB diet may be a more acceptable approach to weight loss than the traditional approach of calorie restriction and mandated exercise with results that potentially improved with time.

Although calorie intake was not assessed, weight loss in this intervention reflects reductions in calorie intake, even with an ad libitum approach, which we hypothesize was due to reduction of dietary energy density and avoidance of ultra-processed foods. There is evidence that reduction of energy density reduces overall energy intake in the short-term [15,16] and aids weight loss in the long-term [17,18]. In short-term ad libitum studies, subjects’ hunger ratings were similar whether they were assigned to low- or high-energy density meals [15,16]. In both a 1-year weight loss study [17] and a separate 6-month evaluation [18], subjects consuming low-energy density diets increased the amount of food they ate while losing more weight than subjects consuming higher-energy density diets. Less hunger while consuming a larger amount of food was consistent with our experience: while we did not quantify these outcomes, we regularly received this positive feedback from participants. Similarly, a recent randomized controlled trial found that subjects being presented with meals that were ultra-processed, as defined by the NOVA classification system [19], with non-beverage foods that were higher in energy density, resulted in an additional 500 Kcal/day consumed compared to an unprocessed diet [20].

Vegetarian and vegan participants in our intervention experienced statistically significant weight loss, with reductions in total and LDL cholesterol comparable to the non-vegetarian participants. Despite the general perception of healthfulness, vegetarian and vegan diets do not exclude processed foods and as a consequence, do not differ much in macronutrient composition from non-vegetarian diets [21]. In the Adventist-Health-Study-2, the percent calories from fat in vegetarian (lacto ovo vegetarian) and vegan (strict vegetarian) diets did not differ significantly from non-vegetarian diets. Strict vegetarians consumed a mean of 29.8% of their calories from fat and lacto ovo vegetarians consumed 33.1% of their calories from fat compared to 35.1% in non-vegetarians [21]. By contrast, the plant-based diets shown to reverse heart disease contained approximately 10% calories from fat [6,7]. This very low-fat content was achieved by exclusion of added pure fats, processed foods, and high-fat plant foods. Our participants were encouraged to consume a similar diet. This likely resulted in a reduction of dietary energy density and subsequent calorie consumption when compared to baseline vegetarian and vegan diets.

There are many other differences between the diet recommended in our intervention and the typical American diet and, to a lesser extent, vegetarian and vegan diets. Each of these differences reflects a number of possible mechanisms by which this intervention may yield improved outcomes. In addition to reductions in dietary energy density and total fat intake, including saturated fat intake, a whole-food plant-based diet contains far less refined carbohydrate in the form of added sugars and refined grains than does the typical American diet. There is likely a lower omega-6 to omega-3 fatty acid ratio in a WFPB diet due to the avoidance of added oils. A WFPB diet contains more fruits, vegetables, and legumes and, consequently, contains more fiber and antioxidants [22]. Simultaneously, of course, those adhering to a WFPB diet consume far less or no meat and dairy and, consequently, consume far less of the nutrients they contain, including animal-based protein. Thus, it is not possible to assume that the health effects of our intervention were due to the changes in intake of any one nutrient or food. While it is academically appealing to isolate the effects of single nutrients on health outcomes to elucidate mechanistic details of how this intervention works, our program was purely intended to be a practical clinical program of behavior change. Our goal was to help patients make as many healthful nutritional changes as possible all at once, utilizing a wide array of potentially synergistic mechanisms to maximize health.

### 4.2. Strengths and Limitations

Limitations of this intervention include its short-term nature, a non-randomized design, and lack of dietary analysis. We did not formally follow participants for over nine weeks, so we do not know if participants were able to maintain their weight loss or other health benefits long-term. We cannot isolate the effects of the diet from other program components (namely, substantial and frequent contact with the study physicians and group peers) in a non-randomized design. We did not perform dietary analysis, so we were not able to characterize the nutrient composition of participants’ baseline diets or their degree of adherence to a WFPB diet. We did not ascertain the duration of vegetarian diet or the type of vegetarian diet in participants who identified as vegetarian or vegan at baseline. It is possible that duration and diet type impacted these participants outcomes. Lipid results were not measured in a fasting state, which limits, in particular, the interpretation of triglyceride results. In addition, lack of standardized lipid measurements, particularly in our first several cohorts, resulted in a smaller sample of measured participants compared to total participants, which could introduce unintentional bias. However, the size of the effect we observed is consistent with other plant-based interventions [23]. Strengths of the study include excellent retention of participants for the duration of the intervention despite using what is routinely considered to be a strict nutritional plan and a real world setting with participants responsible for their own food preparation. Participants in this intervention were highly motivated individuals with the means to afford an intensive self-pay program and are therefore not representative of the general population.

## 5. Conclusions

Our findings suggest that a group program in a primary care setting, utilizing an ad libitum whole-food plant-based diet without calorie counting or portion control, mandated exercise or stress management, resulted in short-term benefits, including weight loss and reductions in blood pressure and blood cholesterol in highly motivated participants, including those who were already vegetarian or vegan.

## Figures and Tables

**Table 1 nutrients-11-02068-t001:** Ad libitum whole-food plant-based diet.

**Allowed foods** Non-starchy vegetables Starchy vegetables (potatoes, legumes) Fruits (whole fresh or frozen, not dried, juiced, or blended) Whole grains Plant-based or non-caloric beverages (unsweetened soy milk or nut beverages, water, green tea, decaffeinated coffee and tea) All culinary spices and herbs Seeds rich in omega-3 fats (ground flaxseed, chia seed)
**Excluded foods** All animal products: Meat, poultry, fish, and seafood; eggs; dairy products Refined flours All added edible oils and solid fats Vegan meat and cheese replacement foods containing added oils Sweets (candy, granola bars, cookies, cakes, and pastries) Caloric or artificially-sweetened beverages (including 100% fruit juices and smoothies)
**Limited foods (*consume sparingly, if at all*)** High-fat plant foods: Raw or dry-roasted nuts; seeds (other than above); coconut; avocado Dried fruits Added sweeteners (including natural or less refined sweeteners) Refined soy or wheat-gluten foods (tofu, isolated soy protein, seitan) Alcoholic beverages (ideally 2 drinks/week or less) Caffeinated beverages

**Table 2 nutrients-11-02068-t002:** Participant characteristics at baseline.

	***n =* 79 (Except as Noted)**
**Sex, n (%)**	
Female	53 (67%)
Male	26 (33%)
**Age, years (SD)**	59.2 (10.7)
**Race and ethnicity, *n* (%)**	
Ethnicity	
Hispanic	2 (3%)
Non-Hispanic	68 (86%)
Did not disclose	9 (11%)
Race	
White	67 (85%)
African American	1 (1%)
American Indian	1 (1%)
Multiracial	1 (1%)
Did not disclose	9 (11%)
**Weight status, *n* (%)**	
Underweight (BMI < 18.5)	1 (1%)
Normal weight (BMI 18.5–24.9)	10 (13%)
Overweight (BMI 25–29.9)	20 (25%)
Obese (BMI ≥ 30)	48 (61%)
**Diagnoses, *n* (%)**	
Prediabetes	8 (10%)
Type 2 diabetes	20 (25%)
Coronary artery disease	6 (8%)
Hypertension	37 (47%)
Dyslipidemia	48 (61%)
History of cancer	8 (10%)
**Statin use, *n* (%)**	24 (30%)
**Dietary pattern, *n* (%)**	
Vegetarian or vegan	24 (30%)
Non-vegetarian	53 (67%)
Unknown	2 (3%)
**Weight, kg (SD)**	92.7 (22.9)
**BMI, kg/m^2^ (SD)**	33.3 (8.9)
**Cholesterol, mg/dL (SD)**	***n =* 73**
Total	179.9 (42.5)
LDL	99.1 (36.0)
HDL	50.3 (14.9)
Triglycerides	152.8 (79.5)
**Blood Pressure, mm Hg (SD)**	***n =* 77**
Systolic BP	137.9 (16.3)
Diastolic BP	85.9 (11.3)

**Table 3 nutrients-11-02068-t003:** Clinical outcomes (average duration of 8.9 ± 1.2 weeks). Data are means ± SD.

	*n* ^1^	Baseline	Final	Change	*p*-Value
BMI, kg/m^2^	78	33.1 (8.8)	31.2 (8.2)	−2.0 (1.1)	<0.0001
Weight, kg	78	92.3 (22.8)	86.8 (20.9)	−5.5 (3.0)	<0.0001
Percent body weight loss, %					
All participants	78			−5.7 (2.7)	
Underweight (BMI < 18.5)	1			−1.7	
Normal weight (BMI 18.5–24.9)	10			−3.0 (2.1)	*
Overweight (BMI 25–29.9)	20			−5.8 (2.8)	
Obese (BMI ≥ 30)	47			−6.4 (2.5)	
Systolic BP, mm Hg	75	137.6 (15.6)	130.5 (17.9)	−7.1 (15.5)	0.0002
Diastolic BP, mm Hg	75	85.9 (11.2)	78.6 (10.5)	−7.3 (10.9)	<0.0001
Cholesterol, mg/dL					
Total	65	180.4 (44.6)	155.2 (40.8)	−25.2 (24.7)	<0.0001
LDL	63	100.3 (38.1)	85.0 (35.3)	−15.3 (21.1)	<0.0001
HDL	65	49.1 (14.9)	41.9 (12.1)	−7.3 (7.8)	<0.0001
Triglycerides	65	155.9 (82.3)	145.7 (72.1)	−10.2 (63.2)	0.198

^1^ Data were not available for all participants. * Overweight and obese groups did not differ significantly from one another in percent body weight loss. Each had a statistically significant difference in percent body weight loss compared to the normal weight group (*p* < 0.05).

**Table 4 nutrients-11-02068-t004:** Outcomes by vegetarian or vegan versus non-vegetarian status at baseline. Data are means ± SD.

	Vegetarian or Vegan				Non-Vegetarian					
	*n* ^1^	Baseline	Final	Change	*n*	Baseline	Final	Change	EFFECT SIZE	*p*
BMI, kg/m^2^	24	32.1 (8.9)	30.5 (8.3)	−1.6 (1.1) ****	52	33.8 (9.0)	31.6 (8.3)	−2.1 (1.1) ****	0.5 (1.1)	0.0711
Weight, kg	24	88.9 (22.1)	84.4 (20.6)	−4.5 (2.8) ****	52	94.2 (23.3)	88.2 (21.4)	−6.0 (3.1) ****	1.5 (3.0)	0.0505
Percent body weight loss, %	24			−4.9 (2.8)	52			−6.1 (2.6)	1.3 (2.7)	0.0642
SBP, mm Hg	23	134.3 (14.3)	130.1 (14.3)	−4.1 (13.3)	50	139.7 (16.0)	130.6 (19.5)	−9.1 (16.3) ***	4.9 (15.4)	0.2090
DBP, mm Hg	23	84.6 (12.9)	79.3 (10.4)	−5.3 (12.3) ^2^	50	86.7 (10.1)	77.8 (10.4)	−8.9 (9.6) ****	3.7 (10.5)	0.1722
Cholesterol, mg/dL										
Total	19	177.2 (38.1)	159.4 (28.0)	−17.7 (18.3) ***	44	182.2 (48.2)	153.3 (45.9)	−28.8 (26.7) ****	11.1 (24.5)	0.1045
LDL	18	103.8 (30.4)	88.3 (22.8)	−15.6 (16.2) ***	43	99.0 (42.0)	83.3 (40.1)	−15.7 (23.2) ****	0.2 (21.4)	0.9751
HDL	19	46.7 (11.9)	42.5 (11.0)	−4.2 (7.0) *	44	50.3 (16.2)	41.5 (12.7)	−8.8 (8.0) ****	4.6 (7.7)	0.0336
Triglycerides	19	134.0 (56.6)	149.3 (100.9)	15.3 (74.9)	44	165.6 (91.8)	146.0 (58.3)	−19.6 (56.2) *	34.9 (62.3)	0.0456

^1^ Data were not available for all participants. **** *p* < 0.0001, *** *p* < 0.001, * *p* < 0.05. ^2^ Approached significance, *p* = 0.0519.

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
