# Peer review of "Evaluation of an Eight-Week Whole-Food Plant-Based Lifestyle Modification Program"

_nutrients, 2019, doi:10.3390/nu11092068_

Round 1

Reviewer 1 Report

The manuscript presented by Campbell and collaborators is an interesting study. The manuscript is well written and structured. The methodology used is adequate and updated. The results with robust and allow to sustain the discussion. However, I have comments that could help improve the understanding and impact of the study.

Major Comments

1. The authors have data on energy, fat, carbohydrates and protein intake ?. This information would allow a better analysis and discussion of the results.

2. Regarding intake, it would be very good to have data on the intake of saturated fat and sugar, in addition to what type of cereals (refined or not) or oils consumed (high content of MUFA and antioxidants).

3. The introduction is well structured, however it is important to indicate that the greatest evidence on benefits of a diet and disease prevention corresponds to the Mediterranean diet (diet that has a high content of dietary fiber, natural antioxidants, MUFA and n-3 PUFA).

Suggested reference:

Serra-Majem et al., Benefits of the Mediterranean diet: Epidemiological and molecular aspects. Mol Aspects Med. 2019; 67: 1-55.

4. In the discussion, it would be interesting for the authors to include a paragraph that the western diet has a high consumption of n-6 PUFA, and a low consumption of n-3 PUFA. Very characteristic aspect in obesity and the further development of NAFLD.

5. In addition, regarding the consumption of oils, it is very likely that the benefits of diets with high plant (vegetable) content are characterized by including extra virgin olive oil (high content of MUFA "oleic acid" and natural antioxidants "tocopherols" and polyphenols ").

Suggested References:
Valenzuela and Videla. The importance of the long-chain polyunsaturated fatty acid n-6 ​​/ n-3 ratio in development of non-alcoholic fatty liver associated with obesity. Food Funct 2011; 2 (11): 644-8.

Hernandez-Rodas et al., Relevant Aspects of Nutritional and Dietary Interventions in Non-Alcoholic Fatty Liver Disease. Int J Mol Sci. 2015; 16 (10): 25168-98.

Minor comments:

1. Improve the wording of the study objective
2. Correct some paragraphs (very long sentences)

Reviewer 2 Report

This was an interesting study that focused on a short-term whole based eating approach. There were a few areas the reviewer suggest to strengthen the manuscript:

Abstract: Recommend either poor diet quality or poor nutritious diet as a poor diet is open to interpretation for the start of this abstract. Recommend a different word than ‘heavier’ as that is an opinion, so those with a higher BMI at start (>30 mg/kg). In the conclusion, it appears those who were non-vegetarians benefits more than those who were vegetarians/vegans, according to your results, thus revise the statement. Also, since this was an 8-week intervention, these are short-term results, thus indicate that in the conclusion as long-term these effects are not seen as much and also individuals resume their normal dietary habits.

Introduction:

In the first sentence, include a citation that specifically links a poor diet to death as the chronic diseases listed may have been related to diet, but it could also be related to overall lifestyle behaviors (exercise, smoking, stress, etc). Quantify a ‘high body mass index’.

Considering the focus of this study was on a plant-based diet, would recommend to include which plant-based diet model was decided for this study or at least expanding on what this type of diet entails as not all plant-based is vegan/vegetarian, it could also include meat occasionally. Also, expand on what aspects of these chronic diseases were reduced and the length of these studies.

Methods:

Either here or in the introduction discuss why the study was 8 weeks as it typically takes 6-months, sometimes shorter, for someone to sustain a behavioral change.

Include the inclusion/exclusion criterion.

Respecting that physicians taught these classes, what nutrition/physical activity and meditation background did they have, especially in regard to behavioral modification strategies? What nutrition background did the chef have?

Quantify the fat amount (ie 20%) and which type of fats were emphasized and how much added sugars as the word minimized was included

For the weight, was it recorded in kg? Also, for those participants who refused to remove their shoes, was weight removed? Please include that information.

Results:

Even though readers may know the BMI for individuals who are considered obese, recommend including that information. For the demographics, was a question asked how long someone was a vegan/vegetarian or what type of vegetarian they were as that may have impacted the biochemistry results. In the methods, it was mentioned that fasting glucose would be measured, but in the results there was no indication of this measurement.

For table 1, include normal weight and underweight, even if no participant fell into these categories.

Discussion:

In the first paragraph, it was difficult to determine your study results to other published results. Suggest your results be first presented to recap the purpose of the study and the main findings and then discuss how your results were similar or not to other studies’.

When discussing about processed vs ultra-processed foods, the definitions of these are not consistent from country to country, therefore expand on what is meant by this. Also, since your study did not appear to collect dietary records, it is difficult to know if your participants consumed processed/ultra-processed foods, so recommend revising this paragraph (lines 203-215).

Recommend including in this discussion about the method in which this intervention was done, which may have also contributed to the results (eg frequent follow up).

Round 2

Reviewer 1 Report

The authors answered all my questions and made the suggested changes.

Reviewer 2 Report

The authors have addressed all the comments from the reviewer. The reviewer has no further comments.